The optimal training intervention for improving the change of direction performance of adolescent team-sport athletes: a systematic review and network meta-analysis

Chen Yonghui
Tulhongjiang Maiwulanjiang
Ling Tianpeng
Feng Xinmiao
Mi Jing taishanmijing@126.com
Liu Ruidong lrd5156@bsu.edu.cn
Sports Coaching College, Beijing Sport University , Beijing, Beijing , China
Chen Yung-Sheng
Electronic publication date: 2025 Feb 21
Publication date: 2025
Volume: 13
Electronic Location ID: e18971
Received 2024 Oct 25; Accepted 2025 Jan 21
Copyright: © 2025 Chen et al.
Copyright year: 2025
Copyright holder: Chen et al.
License: This is an open access article distributed under the terms of the Creative Commons Attribution License, which permits unrestricted use, distribution, reproduction and adaptation in any medium and for any purpose provided that it is properly attributed. For attribution, the original author(s), title, publication source (PeerJ) and either DOI or URL of the article must be cited.
License URL: https://creativecommons.org/licenses/by/4.0/

Keywords: Cutting, Team-sport athletes, Intervention, Youth, Agility

Funding: Key Laboratory of Sport Training of General Administration of Sport of China, Beijing Sport University, Beijing, China the Fundamental Research Funds for the Central Universities 2024YDXL005 This work was supported by the Key Laboratory of Sport Training of General Administration of Sport of China, Beijing Sport University, Beijing 100084, China the Fundamental Research Funds for the Central Universities (NO. 2024YDXL005). The funders had no role in study design, data collection and analysis, decision to publish, or preparation of the manuscript.

==============================
Background

Due to the influence of growth, adolescent team-sport athletes have the need to improve their change of direction (COD) performance and reduce the risk of anterior cruciate ligament (ACL) injuries during COD. However, the optimal intervention for improving COD performance has not yet been determined.

Objective

To quantitatively assess the effects of diverse training interventions on COD performance.

Methods

A systematic search of five databases was conducted, adhering to the PRISMA guidelines. Randomized controlled trials that examined 10 distinct training interventions for COD performance in adolescent team-sport athletes were emphasized. Effect sizes were represented as standardized mean differences (SMD) with 95% credible intervals (CI). The Cochrane study risk assessment tool evaluated the risk of bias in the selected studies.

Results

Of the 36 studies analyzed, involving 1,125 participants. Eccentric overload training (EOT) (SMD = −2.06, 95% CI [−2.83 to −1.29]) emerged as the most effective training method for overall COD performance. Subgroup analysis shows that combined training (COM) (SMD = −2.14, 95% CI [−3.54 to −0.74]) was the best training intervention for COD performance with angles less than 90°. EOT (SMD = −2.84, 95% CI [−4.62 to −1.07]) also was two best training intervention for COD performance with angles greater than 90°.

Conclusions

The choice of training intervention should be determined based on the COD angle. When the COD angle exceeds 90° or is not restricted, EOT is the optimal intervention; however, this is not the case for angles below 90°. Further high-quality studies are needed in the future to validate these findings. Systematic review registration: PROSPERO CRD42024501819.

Introduction

Change of direction (COD) movement is a complex process that involves a sequence of sub-actions, including deceleration, braking, the COD maneuver itself, and acceleration (Brughelli et al., 2008; Chaabene et al., 2018). In many team sports, the ability to perform COD is considered a key factor in determining match outcomes. For example, athletes need to employ COD strategies with various angles to avoid contact with their opponents (Freitas et al., 2022). In a single soccer match, athletes may complete nearly 600 CODs (Machado et al., 2019). Similarly, approximately 60% of playing time in a handball match is spent executing CODs (Póvoas et al., 2012). Elite basketball players perform a change of direction approximately every 1–3 s (Ben Abdelkrim, El Fazaa & El Ati, 2007; Conte et al., 2015). Accordingly, numerous studies have focused on understanding COD and improving COD performance to achieve success in competition (Brughelli et al., 2008; Chia et al., 2021; Donelon et al., 2024; Sariati et al., 2021).

The performance of COD is influenced by various factors. According to the model proposed by Sheppard & Young (2006), these factors include technique, leg muscle qualities, and straight sprinting speed. Strength and power have increasingly become key trainable components of this model. Therefore, to improve competitiveness in matches, fitness trainers and coaches have focused on identifying the most effective training interventions to enhance COD-specific strength and power, thereby improving COD ability.

Additionally, compared to training adults, training youth athletes requires greater caution. Unreasonable intervention choices, such as prolonged exercise duration or excessive training loads, may lead to psychological stress, burnout, and early retirement among youth athletes (Brenner, 2016; Gustafsson, Sagar & Stenling, 2017). Additionally, due to the influence of maturation processes, youth athletes exhibit different COD strategies at various stages of puberty (pre-, mid-, and post-puberty) (Chia et al., 2021; Pardos-Mainer et al., 2020). As youth athletes mature, they tend to adopt strategies characterized by quadriceps dominance (e.g., reduced sagittal-plane hip and knee range of motion) and ligament dominance (e.g., greater peak knee abduction angles) (Chia et al., 2021). These strategies increase the risk of anterior cruciate ligament (ACL) injuries (Chia et al., 2021).

During COD, there is a trade-off between angle and velocity, where faster approaches may compromise the performance of the intended COD (Dos’Santos et al., 2018). Specifically, to perform COD tests involving larger angles, participants must employ strategies such as deceleration in advance or adjusting the angles of COD (Schot, Dart & Schuh, 1995; Suzuki et al., 2014). Otherwise, the knee joint will experience significant load, thereby increasing the risk of ACL injury. Numerous studies have examined the effects of COD at varying angles (Besier et al., 2001; Hader et al., 2016; Havens & Sigward, 2015b) and found a significant increase in the knee valgus moment as the angle progresses from 45° to 90° (Cortes, Onate & Van Lunen, 2011; Hader, Palazzi & Buchheit, 2015; Havens & Sigward, 2015a; Schreurs, Benjaminse & Lemmink, 2017), indicating a greater risk of ACL injury. The lower limb load increases with the COD angle, but this does not seem to be linear (Havens & Sigward, 2015b). Beyond 90°, the increase in valgus moment tends to plateau, with subsequent changes in the moment stabilizing (Dos’Santos et al., 2018; Schreurs, Benjaminse & Lemmink, 2017). This suggests that 90° may represent a potential inflection point, with COD beyond 90° indicating a larger strain on the ACL (Schreurs, Benjaminse & Lemmink, 2017). Fortunately, physical capacity, including the ability to rapidly generate force and maintain neuromuscular control, can help mitigate this trade-off, balancing the performance and injury risk (Dos’Santos et al., 2018). Therefore, to mitigate the rising risk of ACL injuries with age progression or improve the performance of COD, optimal training interventions must be developed and implemented (Dos’Santos et al., 2018; Pardos-Mainer et al., 2020).

However, there is no consensus regarding the optimal training intervention for improving COD performance. Early perspectives suggested that youth athletes should engage in power-based training, such as plyometric training (PT) (Barrera-Domínguez et al., 2023; Malisoux et al., 2006) or change of direction training (CODT) (Dos’Santos et al., 2019a; de Villarreal et al., 2021), rather than traditional resistance training (TRT) (Hoffman et al., 2004, 2005), to enhance COD performance. For example, Markovic (2007) found a small relationship (r = −0.17 to −0.31) between the one-repetition maximum (1RM) isoinertial squats and COD test performance (Beato et al., 2018). Similarly, Hoffman et al. (2004) reported no improvement in COD performance (T-test) among National Collegiate Athletic Association Division III collegiate football athletes after 15 weeks of strength training. In contrast, a recent meta-analysis emphasized the importance of TRT for improving COD performance (Chaabene et al., 2020). Chaabene et al. (2020) concluded that TRT is more favorable for enhancing COD performance in youth athletes compared to adults.

To improve COD performance in adolescent team-sport athletes, a variety of training methods have been employed, including eccentric overload training (EOT) (Allen et al., 2023; Younes-Egana et al., 2023), PT (Asadi et al., 2016; Martín-Moya & González Fernández, 2022; Ramirez-Campillo et al., 2022), and TRT (Biel et al., 2023), among others. However, due to methodological limitations, experimental studies and pairwise meta-analyses have struggled to identify the most effective training intervention for improving COD performance. Furthermore, although multiple meta-analyses have demonstrated the effectiveness of specific training interventions, they have overlooked the critical variable of COD angle (Asadi et al., 2016; Morris et al., 2022; Pardos-Mainer et al., 2021; Ramirez-Campillo et al., 2022). This oversight may result in gaps when applied in practical settings.

In this context, training interventions should be both specific and effective, with the selection of the optimal intervention being angle- or velocity-dependent. With this in mind, the first aim of this network meta-analysis (NMA) is to identify the most effective training intervention for improving COD performance, while the second aim is to propose effective interventions for enhancing COD performance at specific angles.

Materials and Methods

This systematic review and NMA was conducted in accordance with the Preferred Reporting Items for Systematic Reviews and Meta-analyses (PRISMA) statement (PROSPERO ID: CRD42024501819).

Search strategy

A systematic search was conducted in PubMed, Scopus, Cochrane Library, SPORTDiscus, China National Knowledge Infrastructure (CNKI), and Web of Science from their inception dates to December 28, 2024. The keywords used for the search primarily include: (1) training interventions: “resistance training”, “endurance training”, “plyometric training”, “high-intensity interval training”, “HIIT”, “eccentric overload training”, “balance training”, “change of direction training”, “combined training”, “concurrent training”, “complex training”; (2) outcomes: “change of direction performance”, “change of direction ability”, “agility”, “cut*”; population: “adolescen*”, “team-sport athlete*”, “youth athlete*”, “younger player”, “teenage athlete*”, “soccer player*”, “basketball player*”, “rugby player*”, “handball player*”. The specific search strategies, including search terms, dates, and process, are shown in File S1. The reference lists of relevant articles and reviews were also screened for additional studies. Title/abstract and full-text screening were conducted independently and in duplicate by the investigators (Y.H.C. and X.M.F.), with disagreements resolved by discussion or adjudication by the third author (J.M.).

Included criteria

The inclusion criteria were based on the PICOS (participants, interventions, comparators, outcomes, and study design) approach (Hutton et al., 2015). The criteria were as follows: 1) peer-reviewed English or Chinese journal articles (core journals); 2) studies including adolescents aged 11–18, as this age range encompasses all the characteristics of adolescence; 3) experimental studies; and 4) studies related to COD performance. The control group consisted of different exercise interventions or active controls (performing a standard training regimen). The outcome measures of the included studies had to be related to COD performance and expressed in units of time.

Excluded criteria

1. Due to insufficient data availability, reviews, commentaries, case reports, conference abstracts, and similar sources were excluded.

2. Injured adolescent team-sport athletes were excluded.

3. As this study solely investigates the effects of training interventions on COD performance, COD tests involving cognitive engagement will be excluded.

Data extraction

A nine-item and standardized data extraction form was used to record data from the included studies under the following headings: (i) author, (ii) year of publication, (iii) sample size, (iv) event, (v) sample mean physical characteristics, (vi) training frequency, (vii) training duration, (viii) training intensity, (ix) details of training interventions, (x) following up COD performance (e.g., T test and/or Illinois test), (xi) main study results, (xii) study finding. Data extraction was independently conducted by two authors, M.T and T.L.

Risk of bias of individual studies

Two authors independently assessed the risk of bias (ROB). The Cochrane study risk assessment tool was employed to assess the methodology level of studies fulfilling the inclusion criteria. “+” denotes that the studies reach the requirement, while “−” denotes otherwise. “?” denotes that the study did not mention. Two assessors independently rated the studies. Conflicts between ratings were addressed through discussion, and a third assessor was consulted if no consensus was reached.

Data analysis

All statistical analysis was performed using R (version 4.3.2). Since all of included outcomes were measured by different rating instruments, so standardized mean difference (SMD) was used to present the effect size (Andrade, 2020). To assess the efficacy of the interventions, the P-score was calculated to quantify the ranking of intervention measures (Rosenberger et al., 2021). The higher the P-score, the better the disguised performance of the intervention measures. League tables were employed to present the results of direct and mixed comparisons. Statistical heterogeneity between studies was quantified using the Tau square (τ2) test and I2. The larger the τ2 and the smaller the p-value, the greater the possibility of heterogeneity. The I2 statistics was used as a measure of heterogeneity between studies, ranging from 0–100%. The I2 values ranged from 0–100% and were interpreted as follows: <25% low heterogeneity, 25–50% moderate heterogeneity and >75% high heterogeneity.

The side-splitting approach was used to estimate the inconsistency of results between direct and indirect evidence. The p-value of side-splitting approach less than 0.05 indicates a significant difference between direct and indirect comparisons, suggesting that the conclusions may not be robust. Additionally, previous studies have referred to 90° as a cutoff point. If the angles of the COD test exceed 90°, the knee valgus moment increases, the leg load increases (Dos’Santos et al., 2018; Havens & Sigward, 2015b; Young, McDowell & Scarlett, 2001), and the changes in leg load tend to stabilize at higher angles of COD (Schreurs, Benjaminse & Lemmink, 2017). Thus, the 90° was used as a cutoff point to divide the subgroups into those with a COD angle less than 90° (excluding 90°) and those with a COD angle greater than 90° (including 90°). Network funnel plots were generated to assess the presence of bias. In cases where a study included multiple intervention groups, the sample size of the common control group was divided by the number of comparisons made (Arnup et al., 2016).

Results

Descriptions of included studies

A total of 2,770 publication were searched. After excluding 958 reports based on their titles and abstracts, 1,111 references were retrieved for a full inspection. Finally, 36 studies were included (Fig. 1). Ten interventions were included in this study. The definitions and abbreviations of various training interventions are presented in Table 1.

Figure 1 Study selection flowchart.

Table 1 Different training modalities definition and abbreviation.

Abbreviation	Training modality	Definition	
HIIT	High intensity interval training	>80% maximal heart rate or >100% lactate threshold or >90% maximal oxygen uptake (VO2max) and repeated bouts of ≤5 min of exercise (Eddolls et al., 2017).	
PT	Plyometric training	Plyometric training was defined as an exercise with body weight and/or ≤20% of 1RM performed by utilizing the SSC (Wilk et al., 1993).	
CODT	Change of direction training	A training method that uses direction change as a means of practice (Chtara et al., 2017).	
RST	Repeated sprint training	Produce the best possible average sprint performance over a series of sprints (≤10 s), separated by short (≤60 s) recovery periods (Bishop, Girard & Mendez-Villanueva, 2011).	
EOT	Eccentric overload training	Provide additional load or increase the duration of the eccentric stage (e.g., eccentric contraction for 3 s and concentric contraction for 1 s) (Vogt & Hoppeler, 2014).	
CT	Complex training	A combination of traditional resistance training and plyometric training with similar biomechanical characteristics, using post activation potentiation to improve the performance of plyometric training (Hammami et al., 2019c).	
TMT	Trunk muscle training	Training for the trunk or pelvic region with the goal of improving trunk stability (Van Criekinge et al., 2021).	
TRT	Traditional resistance training	By using limbs to resist heavy resistance equipment to enhance the cross-sectional area of muscle fibers and muscle strength (Schoenfeld, 2013).	
COM	Combined training	Perform two different training interventions within a session (e.g., strength training 10 min after endurance training) (Bouteraa et al., 2020).	
INT	Integrated neuromuscular training	Refers to a comprehensive training that combines general basic functional actions with specific strength, plyometric, speed, agility, and balance (Cavaggioni et al., 2024).	
Note:

1RM, one repetition maximum.

The characteristics of 36 studies are summarized in Table 2. The 36 studies with 1,143 participants, 933 (81.63%) of whom were male, and 221 (18.37%) of whom were female. Sample size range from 6 to 25 participants, and age range from 11 to 17 years. Training frequency range from 1 to 3 times per week, training duration range from 4 to 12 weeks. Six studies assessed COD tests with angles under 90°, 25 studies assessed COD tests with angles from 90° to 180° and 12 studies assessed COD tests with angles above and below 90° (i.e., Illinois test). Additionally, a total of 11 studies employed a 3-arm design, while four studies employed a 4-arm design. So, the sample size of the ‘shared’ control group was divided by the number of comparisons.

Table 2 Included studies characteristics.

Author/Year	Intervention characteristics	Participants characteristics	Frequency
(days/week)	Duration
(weeks)	Outcomes (angles of direction × reps, distance)	
Num	M (%)	Height (cm)	Weight (kg)	Age (years)	Event	
Beato et al. (2018)	COM (CODT-PT)	11	100%	175.2 ± 5.9	69.2 ± 6.1	17.0 ± 0.8	Soccer	2	6	5-0-5 (180° × 2, 10 m)	
	CODT	10	100%	178.6 ± 6.5	71.3 ± 6.8	17.0 ± 0.8				
Bouteraa et al. (2020)	COM (BT-PT)

BT: 3 acts × 6–10 sets × 20–30 reps

PT: 3 acts × 2 sets × 1–15 reps

	16	0%	168.0 ± 5.0	56.6 ± 8.3	16.4 ± 0.5	Basketball	2	8	Modified Illinois test (exceeding 60° × 5 and below 60° × 4, ~20 m)	
	Control	10	0%	168.0 ± 8.0	55.6 ± 7.0	16.5 ± 0.5					
Bourgeois et al. (2017)	EOT: 5–8 acts × 3 sets × 6–10 reps

Intensity: 6–10RM

	12	100%	180.0 ± 10.0	81.8 ± 12.4	15.3 ± 0.5	Rugby	3	6	Modified 5-0-5 (180° × 2, 10 m) Specific COD test (45° × 1, 8 m)	
	TRT: 5-8 acts × 3 sets × 6–10 reps

Intensity: 6–10RM

	6	100%	180.0 ± 10.0	81.8 ± 12.4	15.3 ± 0.5				
Chtara et al. (2017)	PT: 6 acts × 2–3 sets × 8–12 reps	10	100%	165.0 ± 7.0	54.1 ± 6.5	13.6 ± 0.3	Soccer	2	7	zigzag test
(100° × 3, 20 m)	
	CODT: 4 acts × 2 sets × 2–4 reps	10	100%	165.0 ± 7.0	54.1 ± 6.5	13.6 ± 0.3		
	RST: 5 acts × 3 sets × 20 m

Intensity: 100% VO2max

	12	100%	165.0 ± 7.0	54.1 ± 6.5	13.6 ± 0.3		
	Control	10	100%	165.0 ± 7.0	54.1 ± 6.5	13.6 ± 0.3		
Dos’Santos et al. (2019a)	CODT	13	100%	177 ± 5	69.2 ± 9.2	16.9 ± 0.2	Soccer	2	6	Specific COD test
(70° × 1, 10 m)	
	Control	13	100%	177 ± 7	73.3 ± 8.1	17.8 ± 0.3					
Fiorilli et al. (2020)	PT: 4 acts × 3–4 sets × 7–10 reps	16	100%	168.0 ± 7.0	52.1 ± 5.2	13.4 ± 0.8	Soccer	2	6	Illinois test
(exceeding 90° × 5 and below 90° × 4, ~60 m)	
	EOT: 2 acts × 4 sets × 7 reps	18	100%	165.0 ± 10.0	51.2 ± 6.7	13.2 ± 1.2		
Gaamouri et al. (2023)	TRT: 6 acts × 3–5 sets × 10 reps

Intensity: 8–16 kg

	17	0%	169.0 ± 4.2	63.4 ± 3.8	15.7 ± 0.2	Handball	2	10	Modified T test
(90° × 2, 180° × 2, 20 m)	
	Control	17	0%	167.0 ± 3.5	63.0 ± 3.8	15.8 ± 0.2		
Genc, Cigerci & Sever (2019)	TMT: 6 acts × 2 sets × 30 reps	10	0%	153.2 ± 5.8	60.0 ± 7.9	17.8 ± 1.4	Handball	3	8	Pro test
(90° × 1, 180° × 2, ~18 m)
5-0-5 test (180° × 2, 10 m)	
	Control	10	0%	166.6 ± 5.8	63.3 ± 6.3	17.6 ± 1.9		
Hammami et al. (2016)	PT: 4 acts × 4–10 sets × 7–10 reps	15	100%	176.0 ± 6.0	59.0 ± 6.5	15.7 ± 0.2	Soccer	2	8	S4 × 5
(90° × 2, and 180° × 1, 20 m)
S180 (180° × 4, 30 m)
RCOD (100° × 4, 20 m)	
	Control	13	100%	169.0 ± 5.0	58.2 ± 5.0	15.8 ± 0.2		
Hammami et al. (2017)	CT:

TRT: 1 act × 3–5 sets × 3–8 reps

Intensity: 70–90% 1RM

PT: 1 act × 3–5 sets × 3–8 reps

	16	100%	178.0 ± 5.0	59.3 ± 6.5	16.0 ± 0.5	Soccer	2	8	S180 (180° × 4, 30 m)
S4 × 5
(90° × 2, and 180° × 1, 20 m)
RCOD (100° × 4, 20 m)	
	TRT: 1 act × 3–5 sets × 3–8 reps

Intensity: 70–90% 1RM

	16	100%	175.0 ± 3.0	58.0 ± 6.2	16.2 ± 0.6		
	Control	12	100%	168.0 ± 5.0	58.1 ± 5.2	16.8 ± 0.2		
Hammami et al. (2018)	TRT: 1 act × 3–5 sets × 3–8 reps

Intensity: 70–90% 1RM

	19	100%	175.0 ± 3.0	58.1 ± 7.3	16.2 ± 0.6	Soccer	2	8	S180 (180° × 4, 30 m)
S4 × 5
(90° × 2, and 180° × 1, 20 m)
RCOD (100° × 4, 20 m)	
	Control	12	100%	168.0 ± 5.0	58.2 ± 5.0	15.8 ± 0.2		
Hammami et al. (2019b)	COM (CODT-PT)

PT: 4 acts × 2 sets × 6 reps

RST: 4 acts × 2 sets × 3–6 reps

	14	100%	178.0 ± 5.0	69.3 ± 3.1	14.5 ± 0.3	Handball	2	8	Modified T test
(90° × 2, 180° × 2, 20 m)
Modified Illinois test
(exceeding 90° × 5 and below 90° × 4, ~20 m)	
	Control	14	100%	172.0 ± 7.0	66.9 ± 4.7	14.6 ± 0.2					
Hammami et al. (2019a)	CT:

TRT: 1 act × 4 sets × 1 × 6 reps

Intensity: 75– 85% 1RM

PT: 1 act × 4–10 sets × 7–10 reps

	14	0%	163.0 ± 4.0	60.8 ± 4.7	16.6 ± 0.3	Soccer	2	8	S4 × 5
(90° × 2, and 180° × 1, 20 m)	
	PT: 1 act × 4–10 sets × 7–10 reps	14	0%	175.0 ± 6.0	58.9 ± 6.7	15.7 ± 0.2					
	Control	12	0%	168.0 ± 5.0	58.3 ± 5.2	15.8 ± 0.2					
Hammami et al. (2021)	HIIT: 2 acts × 4 sets × 6 reps	17	100%	178.0 ± 3.0	69.0 ± 5.4	16.6 ± 0.5	Handball	2	8	Modified T test
(90° × 2, 180° × 2, 20 m)
Modified Illinois test
(exceeding 90° × 5 and below 90° × 4, ~20 m)	
	Control	15	100%	176.0 ± 7.0	68.9 ± 5.3	16.5 ± 0.8				
Haghighi et al. (2023)	PT: 4 acts × 2–3 sets × 4–8 reps	8	0%	168.3 ± 8.7	61.7 ± 10.3	14.6 ± 1.5	Basketball	3	6	Lane test
(90° × 6, 180° × 1, ~42 m)	
	HIIT: 10 acts × 1–2 sets × 30–45 s

Intensity: 100% VO2max

	8	0%	167.0 ± 5.5	53.5 ± 3.0	15.1 ± 1.6		
	Control	8	0%	165.8 ± 9.7	56.7 ± 13.6	15.1 ± 1.8		
İnce (2019)	TRT: 3 acts × 3 sets × 5r eps

Intensity: 70–85% 1RM

	17	0%	166.0 ± 5.7	63.4 ± 2.9	15.6 ± 1.3	Volleyball	2	6	T test
(90° × 2, 180° × 2, ~36 m)	
	Control	17	0%	167.5 ± 5.7	60.5 ± 4.1	15.2 ± 1.8		
Keller et al. (2020)	PT: 6 acts × 4–7 sets × 5 reps	12	100%	175.0 ± 11.0	63.0 ± 14.0	14.0 ± 0.8	Team sports	2	4	6 COD test (exceeding 90° × 2 and 180° × 1 and 90° × 2 and below 90° × 1, ~26 m)
13 COD test (exceeding 90° × 4 and 180° × 2 and 90° × 4 and below 90° × 2, ~52 m)
Modified T test
(90° × 2, 180° × 2, 20 m)	
	PT: 6 acts × 3 sets × 8–10 reps	12	100%	175.0 ± 11.0	63.0 ± 14.0	14.0 ± 0.8		
	TRT: 6 acts × 4–6 sets × 4–10 reps

Intensity: until failure

	9	100%	175.0 ± 11.0	63.0 ± 14.0	14.0 ± 0.8		
Maio Alves et al. (2010)	COM (RST-TRT)

TRT: 3 acts × 3 sets × 6 reps

Intensity: 85–90% 1RM

RST: 3 sets × 8 reps

Distance: 5 m

	9	100%	175.3 ± 6.3	70.3 ± 8.3	17.4 ± 0.6	Soccer	2	6	5-0-5 test (180° × 2, 10 m)	
	COM (RST-TRT)

TRT: 3 acts × 3 sets × 6 reps

Intensity: 85–90% 1RM

RST: 3 sets × 8 reps

Distance: 5 m

	8	100%	175.3 ± 6.3	70.3 ± 8.3	17.4 ± 0.6		1	6		
	Control	6	100%	175.3 ± 6.3	70.3 ± 8.3	17.4 ± 0.6					
Makhlouf et al. (2016)	COM (TRT-HIIT)

TRT: 4–5 acts × 3 sets × 5–10 reps

Intensity: free weight

HIIT: 1 act × 2 sets × 12–16 reps

Intensity: 110-120% max speed

	15	100%	164 ± 8.3	53.5 ± 8.6	13.7 ± 0.5	Soccer	2	12	Specific COD test (below 60° × 2 and exceeding 60° × 2)	
	COM (HIIT-TRT)

TRT: 4–5 acts × 3 sets × 5–10 reps

Intensity: free weight

HIIT: 1 act × 2 sets × 12–16 reps

Intensity: 110–120% max speed

	14	100%	164.0 ± 8.3	53.5 ± 8.6	13.7 ± 0.5					
	COM (TRT-HIIT) in alternate day

TRT: 4–5 acts × 3 sets × 5–10 reps

Intensity: free weight

HIIT: 1 act × 2 sets × 12–16 reps

Intensity: 110–120% max speed

	15	100%	164.0 ± 8.3	53.5 ± 8.6	13.7 ± 0.5					
	Control	14	100%	164.0 ± 8.3	53.5 ± 8.6	13.7 ± 0.5					
Makhlouf et al. (2018)	COM (BT-PT)

BT: 5 acts × 1–3 sets × 8–45 reps

PT: 5 acts × 1–2 sets × 8–15 reps

	21	100%	145.4 ± 7.1	36.9 ± 7.7	11.1 ± 0.8	Soccer	2	8	S180 (180° × 4, 30 m)
Illinois test with ball
(exceeding 90° × 5 and below 90° × 4, ~50 m)
Illinois test without ball
(exceeding 90° × 5 and below 90° × 4, ~50 m)	
	COM (AT-PT)

AT: 7 acts × 1 set × 2–8 reps

PT: 5 acts × 1–2 sets × 8–15 reps

	20	100%	147.9 ± 7.4	36.6 ± 8.3	11.3 ± 0.9		
	Control	16	100%	145.3 ± 5.3	37.2 ± 8.0	11.0 ± 0.8		
Mathisen & Danielsen (2014)	COM (CODT-RST)	13	0%	157.6 ± 3.8	52.5 ± 9.3	13.6 ± 0.2	Soccer	1	8	Specific COD test (90° × 2 and 180° × 2, 20 m)	
	Control	13	0%	159.6 ± 5.7	53.8 ± 7.8	13.7 ± 0.3					
Meylan & Malatesta (2009)	PT: 4 acts × 2–4 sets × 6–12 reps	14	100%	159.0 ± 9.0	48.6 ± 9.6	13.3 ± 0.6	Soccer	2	8	Specific COD test (120° × 4, 10 m)	
	Control	11	100%	163.0 ± 10.0	47.4 ± 9.6	13.1 ± 0.6		
Michailidis, Tabouris & Metaxas (2019)	COM (PT-CODT)	17	100%	149 ± 8.6	42.9 ± 6.9	11.8 ± 0.8	Soccer	2	6	T test (90° × 2, 180° × 2, ~36 m)	
	Control	14	100%	152 ± 9.3	43.1 ± 6.4	12.2 ± 0.6				
Negra et al. (2016)	TRT: 4 acts × 4 sets × 8–12 reps

Intensity: 40–60% 1RM

	13	100%	160.4 ± 9.1	49.2 ± 8.1	12.8 ± 0.3	Soccer	3	12	Illinois test (exceeding 60° × 5 and below 60° × 4, ~50 m)
T test (90° × 2, 180° × 2, ~36 m)	
	Control	11	100%	154.5 ± 11.1	45.4 ± 8.1	12.7 ± 0.3		
Negra et al. (2020)	PT: 2 acts × 5–6 sets × 10 reps	13	100%	158.6 ± 4.5	43.7 ± 5.7	12.7 ± 0.2	Soccer	2	8	T test (90° × 2, 180° × 2, ~36 m)	
	Control	11	100%	152.0 ± 6.0	40.0 ± 5.8	12.7 ± 0.2		
Otero-Esquina et al. (2017)	CT:

RT: 2 acts × 2–3 sets × 4–6 reps

Intensity: 40–55% 1RM

PT: 5 acts × 1–4 sets × 3–6 reps

	12	100%	176.7 ± 2.2	69.4 ± 4.2	17.0 ± 1.0	Soccer	2	7	V test (45° × 4, 25 m)	
	CT:

RT: 2 acts × 2–3 sets × 4–6 reps

Intensity: 40–55% 1RM

PT: 5 acts × 1–4 sets × 3–6 reps

	12	100%	176.7 ± 2.2	69.4 ± 4.2	17.0 ± 1.0		1	7	
	Control	12	100%	176.7 ± 2.2	69.4 ± 4.2	17.0 ± 1.0		
Pavillon et al. (2021)	CODT: 4 acts × 1 set × 10 reps	9	100%	165.9 ± 9.2	86.4 ± 3.9	15.9 ± 0.4	Soccer	2	12	Slalom sprint test 5
(45° × 2 and 90° × 2, 20 m)
Slalom sprint test 10
(45° × 2 and 90° × 2, 20 m)	
	CODT: 4 acts ×1 set × 10 reps	9	100%	155.3 ± 7.6	43.2 ± 5.5	13.5 ± 0.8				
	RST: 1 act × 2 sets × 10 reps	9	100%	165.6 ± 6.6	82.7 ± 3.6	16.1 ± 0.6				
	RST: 1 act × 2 sets × 10 reps	10	100%	154.6 ± 7.2	40.5 ± 6.5	13.4 ± 0.6					
Panagoulis et al. (2020)	INT: 4 acts × 3 sets × 5–8 reps	14	100%	163.0 ± 10.0	Not given	12.4 ± 0.6	Soccer	3	8	Arrowhead test (exceeding 90° × 3, 37 m)	
	Control	14	100%	162.0 ± 10.0	Not given	12.2 ± 0.5				
Ramírez-Campillo et al. (2015)	PT: 12 acts × 2–6 sets × 5 reps	12	100%	144.0 ± 17.5	42.2 ± 16.9	11.6 ± 2.7	Soccer	3	6	Specific COD test (120° × 4, 10 m)	
	PT: 6 acts × 6 sets × 5 reps	16	100%	147.0 ± 11.1	45.0 ± 9.3	11.6 ± 1.7				
	PT: 2 acts × 6 sets × 5 reps	12	100%	146.0 ± 13.7	43.5 ± 14.9	11.0 ± 2.0				
	Control	14	100%	143.0 ± 17.7	41.8 ± 12.7	11.2 ± 2.4				
Ramirez-Campillo et al. (2018)	PT: 3 acts × 6–9 sets × 8–10 reps	25	100%	153.0 ± 10.0	46.7 ± 10.5	13.9 ± 1.9	Soccer	2	7	Illinois test (exceeding 90° × 5 and below 60° × 4, ~50 m)	
	PT: 3 acts × 6–9 sets × 8–10 reps	24	100%	153.0 ± 10.0	47.2 ± 11.5	13.1 ± 1.7				
	Control	24	100%	155.0 ± 10.0	49.1 ± 11.1	13.7 ± 1.6				
Ramirez-Campillo et al. (2020)	PT: 3 acts × 5 sets × 8–14 reps	8	100%	159.3 ± 16.7	45.6 ± 13.4	12.1 ± 2.2	Soccer	2	8	Specific COD test
(60° × 4, 10 m)	
	PT: 3 acts × 5 sets × 8–14 reps	8	100%	154.0 ± 11.6	44.4 ± 12.5	12.9 ± 1.9				
	Control	7	100%	155.9 ± 13.0	45.6 ± 10.3	12.6 ± 1.8				
de Villarreal et al. (2015)	COM (RST-PT)

PT: 6 acts × 2 sets × 6 reps

RST: 1 set × 6 reps

Distance: 10 m

	13	100%	168.0 ± 7.8	57.1 ± 8.4	15.3 ± 0.3	Soccer	2	9	Specific COD test
(60° × 4, 10 m)	
	Control	13	100%	165.2 ± 8.5	54.5 ± 6.6	14.9 ± 0.2				
Sanchez-Sanchez et al. (2019)	CODT: 3 acts × 1 set × 10 reps	10	100%	169.7 ± 7.6	62.7 ± 8.8	14.7 ± 0.5	Soccer	2	8	Specific COD test
(exceeding 90° × 3, 70 m)	
	CODT: 3 acts × 1 set × 10 reps	10	100%	169.1 ± 5.8	58.4 ± 6.6	14.4 ± 0.5				
	Control	9	100%	169.1 ± 6.8	62.5 ± 7.1	14.9 ± 0.4				
Shui & Fu (2018)	INT: 7 acts × 2–3 sets × 10–20 reps or 30 s	9	0%	162.5 ± 4.4	48.7 ± 8.0	17.4 ± 1.0	Soccer	3	6	Arrowhead test
(exceeding 90° × 3, 37 m)	
	Control	9	0%	161.9 ± 3.7	49.9 ± 7.1	17.5 ± 0.8				
Tous-Fajardo et al. (2016)	COM (EOT-VT)

EOT: 1 act × 2 sets × 6–8 reps

VT: whole body vibration with 30 Hz and 4-mm amplitude

	14	100%	174.4 ± 6.4	67.6 ± 7.9	17.0 ± 0.5	Soccer	1	11	V test
(60° × 4, 25 m)	
	TRT: 3 acts × 9 sets × 6–10 reps

Intensity: 90% 1RM

	12	100%	174.4 ± 6.4	67.6 ± 7.9	17.0 ± 0.5				
Xu & Lang (2015)	PT: total 80–120 reps	12	100%	180.6 ± 4.5	67.7 ± 7.5	16.4 ± 0.4	Soccer	3	8	Pro test (90° × 1 and 180° × 2, 20 m)
T test (90° × 2 and 180° × 2, 40 m)
Compass test (180° × 3 and 90° × 3, 21 m)
Nebraska test (below 90° × 2 and 90° × 2 and exceeding 90° × 1, 60 m)
AFL test (exceeding 180° × 1, exceeding 90° × 4, 22 m)
Illinois test (exceeding 90° × 5 and below 90° × 4, ~50 m)	
	TRT: 5 acts × 8–15 sets × 3–6 reps

Intensity: 65–75% 1RM

	10	100%	180.6 ± 4.5	67.7 ± 7.5	16.4 ± 0.4				
Note:

Num, number; M, Man; 1RM, one repetition maximum; reps, repetitions; HIIT, high intensity interval training; PT, plyometric training; BT, balance training; CODT, change of direction training; RST, repeated sprint training; EOT, eccentric overload training; CT, complex training; TMT, trunk muscle training; TRT, traditional resistance training; COM, combined training; INT, integrated neuromuscular training; VT, vibration training; AT, agility training.

Risk of bias

Details of the ROB assessment in each study included are provided in File S2. Overall, two studies were judged to be of excellent, 28 studies were judged to be of good, and eight studies were judged to be of poor (File S2).

Network meta-analysis

Figure 2 shows the results of comparing the training interventions with the control group.

Figure 2 Forest plot change in efficacy of primary analysis and subgroup analysis.

(A) change of direction without any angle restriction; (B) change of direction with angles less than 90°; (C) change of direction with angles greater than 90°. The black square represents the estimate of the direct comparison, and the line next to the square indicates the confidence interval (CI) of the direct comparison estimate. Shorter lines signify more precise results. The diamond represents the effect size derived from the combination of direct and indirect comparisons, while the horizontal bar illustrates the range of effect sizes that might be observed in future studies.

Overall COD performance analysis

All available comparisons from the included trials were shown in the network plot for COD performance (Fig. 3).

Figure 3 Network meta-analysis of eligible comparison for change of direction without any direction restriction.

Each node represents an intervention, and the lines between circles represent direct comparisons. The thicker the line, the greater the number of studies conducting direct comparisons.

Apart from TMT (P-score: 0.1010, SMD = −0.03, 95% CI [−1.09 to 1.02]), all other nine types of training interventions significantly improved COD performance, with SMD (95% Credible Interval (CI)) ranging from −1.40 (−2.19 to −0.60) for EOT to −0.79 (−1.13 to −0.44) for TRT (Table 3). Among these, EOT ranked highest in effectiveness. EOT (P-score: 0.8073), CT (P-score: 0.8028), INT (P-score: 0.7626), and HIIT (P-score: 0.7384) demonstrated comparable effectiveness in enhancing COD performance (File S3). The remaining five training interventions also showed significant improvements in COD performance, albeit with comparatively lower effectiveness.

Table 3 League table of primary analysis and subgroup analysis.

Change of direction performance without angle restriction	
EOT	NA	NA	NA	NA	NA	−1.13 (−2.12; −0.13)	NA	0.10 (−1.03; 1.22)	NA	NA	
−0.06 (−0.99; 0.88)	CT	NA	NA	NA	NA	−0.67 (−2.09; 0.76)	NA	−0.92 (−1.73; −0.10)	NA	−1.32 (−1.92; −0.72)	
−0.01 (−1.36; 1.33)	0.04 (−1.17; 1.26)	INT	NA	NA	NA	NA	NA	NA	NA	−1.38 (−2.47; −0.30)	
−0.12 (−1.25; 1.01)	−0.06 (−1.05; 0.92)	−0.11 (−1.47; 1.26)	HIIT	NA	NA	−0.03 (−1.59; 1.52)	NA	NA	NA	−1.28 (−2.14; −0.42)	
−0.50 (−1.64; 0.64)	−0.44 (−1.43; 0.55)	−0.49 (−1.86; 0.88)	−0.38 (−1.55; 0.78)	RST	0.03 (−0.67; 0.72)	−0.60 (−2.08; 0.87)	NA	NA	NA	−1.10 (−2.60; 0.39)	
−0.57 (−1.57; 0.43)	−0.51 (−1.34; 0.32)	−0.56 (−1.81; 0.70)	−0.45 (−1.48; 0.58)	−0.07 (−0.74; 0.60)	CODT	−0.65 (−2.15; 0.85)	−0.08 (−1.56; 1.40)	NA	NA	−0.68 (−1.43; 0.07)	
−0.57 (−1.33; 0.20)	−0.51 (−1.10; 0.08)	−0.55 (−1.68; 0.58)	−0.45 (−1.30; 0.41)	−0.06 (−0.93; 0.80)	0.00 (−0.67; 0.68)	PT	NA	−0.29 (−0.73; 0.15)	NA	−0.75 (−1.11; −0.39)	
−0.60 (−1.46; 0.26)	−0.54 (−1.18; 0.10)	−0.59 (−1.72; 0.55)	−0.48 (−1.37; 0.41)	−0.10 (−0.98; 0.78)	−0.03 (−0.71; 0.65)	−0.03 (−0.48; 0.41)	COM	−1.03 (−2.51; 0.46)	NA	−0.74 (−1.10; −0.39)	
−0.61 (−1.38; 0.16)	−0.55 (−1.14; 0.03)	−0.60 (−1.74; 0.54)	−0.49 (−1.37; 0.39)	−0.11 (−1.00; 0.77)	−0.04 (−0.74; 0.66)	−0.05 (−0.38; 0.29)	−0.01 (−0.48; 0.45)	TRT	NA	−1.15 (−1.62; −0.69)	
−1.36 (−2.69; −0.04)	−1.31 (−2.50; −0.12)	−1.35 (−2.87; 0.16)	−1.25 (−2.58; 0.09)	−0.86 (−2.21; 0.48)	−0.79 (−2.02; 0.43)	−0.80 (−1.90; 0.30)	−0.76 (−1.87; 0.34)	−0.75 (−1.86; 0.36)	TMT	−0.03 (−1.09; 1.02)	
−1.40 (−2.19; −0.60)	−1.34 (−1.89; −0.79)	−1.38 (−2.47; −0.30)	−1.28 (−2.10; −0.46)	−0.90 (−1.73; −0.06)	−0.83 (−1.45; −0.20)	−0.83 (−1.14; −0.52)	−0.80 (−1.13; −0.46)	−0.79 (−1.13; −0.44)	−0.03 (−1.09; 1.02)	CON	
Change of direction performance with angles below 90°	
COM	NA	−0.67 (−1.50; 0.16)	NA	NA	−1.23 (−1.84; −0.63)	NA	
−0.31 (−1.32; 0.71)	CODT	NA	NA	NA	−0.93 (−1.74; −0.11)	NA	
−0.67 (−1.50; 0.16)	−0.36 (−1.67; 0.95)	TRT	NA	NA	NA	−1.47 (−2.60; −0.34)	
−0.81 (−1.76; 0.14)	−0.50 (−1.60; 0.59)	−0.14 (−1.40; 1.12)	PT	NA	−0.42 (−1.16; 0.31)	NA	
−1.01 (−1.85; −0.18)	−0.71 (−1.70; 0.29)	−0.34 (−1.52; 0.83)	−0.21 (−1.13; 0.72)	CT	−0.22 (−0.79; 0.35)	NA	
−1.23 (−1.84; −0.63)	−0.93 (−1.74; −0.11)	−0.56 (−1.59; 0.46)	−0.42 (−1.16; 0.31)	−0.22 (−0.79; 0.35)	CON	NA	
−2.14 (−3.54; −0.74)	−1.84 (−3.56; −0.11)	−1.47 (−2.60; −0.34)	−1.33 (−3.02; 0.36)	−1.13 (−2.76; 0.50)	−0.91 (−2.43; 0.62)	EOT	
Change of direction performance with angles exceeding 90°	
EOT	NA	NA	NA	NA	NA	−1.64 (−3.19; −0.09)	NA	NA	NA	NA	
−0.83 (−2.50; 0.83)	CT	NA	NA	NA	NA	−0.96 (−1.68; −0.23)	NA	−0.67 (−1.95; 0.61)	NA	−1.60 (−2.27; −0.94)	
−0.79 (−2.69; 1.10)	0.04 (−1.13; 1.21)	INT	NA	NA	NA	NA	NA	NA	NA	−1.71 (−2.72; −0.69)	
−1.48 (−3.17; 0.20)	−0.65 (−1.44; 0.14)	−0.69 (−1.83; 0.46)	COM	NA	NA	NA	0.08 (−1.26; 1.42)	NA	NA	−1.07 (−1.63; −0.51)	
−1.44 (−3.55; 0.67)	−0.61 (−2.10; 0.89)	−0.65 (−2.36; 1.06)	0.04 (−1.43; 1.51)	RST	NA	NA	NA	NA	NA	−1.06 (−2.44; 0.31)	
−1.52 (−3.35; 0.31)	−0.69 (−1.75; 0.38)	−0.73 (−2.08; 0.63)	−0.04 (−1.08; 1.00)	−0.08 (−1.72; 1.56)	HIIT	NA	NA	−0.03 (−1.45; 1.39)	NA	−0.95 (−1.91; 0.01)	
−1.64 (−3.19; −0.09)	−0.80 (−1.41; −0.20)	−0.85 (−1.93; 0.24)	−0.16 (−0.81; 0.50)	−0.20 (−1.63; 1.23)	−0.12 (−1.09; 0.86)	TRT	NA	0.42 (−0.94; 1.78)	NA	−0.92 (−1.34; −0.51)	
−1.70 (−3.45; 0.04)	−0.87 (−1.78; 0.04)	−0.91 (−2.14; 0.33)	−0.22 (−1.01; 0.57)	−0.26 (−1.80; 1.28)	−0.18 (−1.31; 0.95)	–0.06 (–0.86; 0.74)	CODT	−0.65 (−2.02; 0.72)	NA	−0.31 (−1.28; 0.66)	
–1.69 (–3.32; –0.07)	–0.86 (–1.51; –0.22)	–0.90 (–1.97; 0.17)	–0.21 (–0.83; 0.41)	–0.25 (–1.67; 1.16)	–0.18 (–1.10; 0.75)	–0.06 (–0.55; 0.43)	0.01 (−0.73; 0.74)	PT	NA	−0.80 (−1.15; −0.44)	
–2.47 (–4.33; –0.60)	–1.64 (–2.76; –0.51)	–1.68 (–3.07; –0.28)	–0.99 (–2.08; 0.11)	–1.03 (–2.70; 0.65)	–0.95 (–2.26; 0.36)	–0.83 (–1.86; 0.20)	−0.77 (−1.95; 0.42)	−0.77 (−1.79; 0.24)	TMT	−0.03 (−0.99; 0.92)	
–2.50 (–4.10; –0.90)	–1.67 (–2.26; –1.08)	–1.71 (–2.72; –0.69)	–1.02 (–1.55; –0.49)	–1.06 (–2.44; 0.31)	–0.98 (–1.88; –0.09)	–0.86 (–1.26; –0.47)	−0.80 (−1.50; −0.10)	−0.81 (−1.14; −0.47)	−0.03 (−0.99; 0.92)	CON	
Note: All results are presented in the form of SMD (95% Crl). The results of the network meta-analysis are showed in the lower left part, and results from direct comparisons in the upper right half (if available). The ranking of different training interventions is based on the P-score, with the effectiveness of interventions decreasing from left to right. Bold indicates significant differences in results. NA: not available.

COD performance analysis with angles less than 90°

Six studies involving 210 participants showed the effect of training for COD performance with angles less than 90°. All available comparisons from the included trials were shown in the network plot for COD performance (Fig. 4).

Figure 4 Network meta-analysis of eligible comparison for change of direction with less than 90°.

Each node represents an intervention, and the lines between circles represent direct comparisons. The thicker the line, the greater the number of studies conducting direct comparisons.

Table 3 and File S3 indicate that COM (P-score = 0.9348, SMD = −1.23, 95% CI [−1.85 to −0.18]) and CODT (P-score = 0.7811, SMD = −0.93, 95% CI [−1.74 to −0.11]) significantly improved COD performance at angles less than 90° compared to active control. In contrast, the other three training interventions (TRT, CT, and EOT) did not demonstrate a significant impact on COD performance at angles less than 90°. Notably, active control (P-score = 0.0493, SMD = −0.06, 95% CI [−1.47 to 1.58]) was found to be more effective than EOT.

COD performance analysis with angles greater than 90°

Twenty-five studies involving 933 participants showed the effect of training for COD performance with angles greater than 90°. All available comparisons from the included trials were shown in the network plot for COD performance (Fig. 5).

Figure 5 Network meta-analysis of eligible comparison for change of direction with greater than 90°.

Each node represents an intervention, and the lines between circles represent direct comparisons. The thicker the line, the greater the number of studies conducting direct comparisons.

Additionally, Table 3 show that EOT (P-score = 0.9371, SMD = −2.50, 95% CI [−4.10 to −0.90]) is the most effective training intervention for improving COD performance with angles greater than 90°. The other seven types of training intervention, including CT (P-score = 0.8224), INT (P-score = 0.8041), COM (P-score = 0.5317), HIIT (P-score = 0.5011), TRT (P-score = 0.4303), CODT (P-score = 0.3989), and PT (P-score = 0.3889), can significantly resulted in improved COD performance with angles greater than 90° following intervention compared with active control. Compared to active control, RST (P-score = 0.5286, SMD = −1.06, 95% CI [−2.44 to 0.31]) and TMT (P-score = 0.1, SMD = −0.03, 95% CI [−0.99 to 0.92]) cannot significantly improve the COD performance with angles greater than 90°.

Heterogeneity, inconsistency and publication bias analysis

The heterogeneity of the analysis of COD tests without angles restriction and COD tests with angles greater than 90° were moderate to high, while that of COD tests with angles less than 90° was low (File S3). The SIDE test revealed no evidence of inconsistency in all analyses, as indicated by P-values ranging from 0.1532 to 0.8808 (File S4). In addition, our comparison-adjusted funnel plot had good symmetry, and the linear fitting line was not perpendicular to quadrant 0. Therefore, no small study effect was found in all analysis and there is no publication bias (File S5).

Discussion

Compared to adult team-sport athletes, adolescent team-sport athletes face rapid increases in height during puberty, which can lead to a decline in COD performance (Chaabene et al., 2020) and increasing the risk of ACL injuries (Chia et al., 2021). Appropriate training interventions can enhance COD performance and reduce the risk of ACL injuries (Acevedo et al., 2014). Although the effectiveness of specific training interventions has been demonstrated (Asadi et al., 2016; Liao et al., 2021; Morris et al., 2022; Oliver et al., 2024; Ramirez-Campillo et al., 2022), no study has yet identified the optimal training program to enhance COD performance specifically in adolescent team-sport athletes. In practice, different training interventions target distinct aspects of performance. For example, PT focuses primarily on the development of neuromuscular control ability (Markovic & Mikulic, 2010), whereas TRT focuses primarily on the development of muscular strength or muscular hypertrophy (Stricker, Faigenbaum & McCambridge, 2020). Therefore, the findings of this review are particularly valuable.

Our study revealed that: 1) EOT was the most effective intervention for enhancing COD performance when the angles of the COD test were not restricted; 2) subgroup analysis using a 90° threshold demonstrated that EOT maintained its effectiveness in COD tests involving angles greater than 90°; and 3) in COD tests with angles less than 90°, the effectiveness of EOT was reduced, with only COM and CODT interventions showing significant improvements in performance. These findings underscore the angle-dependent efficacy of training modalities and provide practical insights for designing targeted interventions to optimize COD performance under varying directional demands.

Recent studies have increasingly emphasized the efficacy of EOT in improving COD performance. Compared to TRT, EOT represents a specialized variation that focuses on the eccentric phase of muscle contraction. This is typically achieved through the use of specialized eccentric training equipment, such as flywheels (Raya-González et al., 2022), or through verbal instructions that emphasize the eccentric phase of movement (Suchomel et al., 2019), inducing significant training stress and physiological strain during practice (Douglas et al., 2017). Metrics related to strength, power, and the stretch-shortening cycle (SSC) are particularly responsive to EOT compared to concentric-only exercises or TRT (Douglas et al., 2017), and these attributes are closely associated with enhanced COD performance. These additional benefits of EOT may stem from multiple factors. First, EOT improves muscle activation. It enhances the excitability of agonist muscles while reducing co-activation of antagonist muscles (Higbie et al., 1996; Hortobágyi et al., 1996; Komi & Buskirk, 1972). Evidence from electromyography (EMG) studies suggests that EOT increases motor unit discharge rates and improves the ability to rapidly recruit larger motor units following long-term training (Vangsgaard et al., 2014). This implies that EOT contributes to improved power output and neuromuscular control (Aagaard, 2003). Second, EOT strengthens the ability to control eccentric contractions (Douglas et al., 2017; Papadopoulos et al., 2014). Following EOT, coordination between agonist and antagonist muscles during the eccentric phase is enhanced, leading to improved SSC efficiency and power performance. For instance, Papadopoulos et al. (2014) reported that after EOT, participants exhibited smaller changes in joint angles at the ankle, knee, and hip during depth jumps, accompanied by greater jump heights, increased explosiveness, and shorter ground contact times (Papadopoulos et al., 2014). Increased joint stiffness indicates better utilization of eccentric contractions, enabling participants to maximize elastic potential energy during the SSC process (Brughelli & Cronin, 2008). These positive adaptations likely reflect the optimization of deceleration, directional change, and subsequent acceleration phases during COD movements following prolonged EOT (Jones et al., 2017).

To further refine the findings of this study, subgroup analyses were conducted by synthesizing studies focusing on COD angles greater than 90°. The results demonstrated that EOT is the most effective interventions, confirming the applicability of our findings for sharper COD angles. The execution of larger-angle COD requires a longer eccentric work duration for the quadriceps, a greater knee flexion angle, and higher knee load (Dos’Santos et al., 2018; Hader et al., 2016; Schreurs, Benjaminse & Lemmink, 2017). As mentioned earlier, higher eccentric strength may enhance eccentric control (Douglas et al., 2017; Papadopoulos et al., 2014), increase joint stiffness (Brughelli & Cronin, 2008), and subsequently improve the SSC (Brughelli & Cronin, 2008), thereby promoting improvements in COD performance. On the other hand, consistent with analyses that did not impose angle restrictions, TMT showed limited efficacy in improving COD performance. Core strength training, a major subtype of TMT, can elicit several specific adaptations, such as enhancing core stability and reducing the risk of ACL injuries during cutting maneuvers (Alentorn-Geli et al., 2009). While these adaptations have been shown to correlate with COD performance, improvements in lower-limb strength and power may play a more critical role in enhancing COD ability. This likely explains the relatively modest improvements associated with TMT. However, research by Dos’Santos et al. (2019b) suggests that TMT can effectively reduce the risk of ACL injuries during cutting movements. Given the substantial ACL injury risk inherent to COD actions, incorporating this type of training intervention into the routines of adolescent team-sport athletes is a prudent approach. Although increases in strength and power can also mitigate ACL injury risk to some extent, these adaptations may simultaneously enable athletes to execute higher sprinting speeds and sharper COD angles, potentially elevating the risk of ACL injuries (Hader, Palazzi & Buchheit, 2015; Vanrenterghem et al., 2012). Therefore, we propose that TMT offers a safer and more cautious approach to reducing ACL injury risk in adolescent team-sport athletes.

While the superiority of EOT in enhancing COD performance among adolescent team-sport athletes has been demonstrated in both unrestricted COD tests and those with angles greater than 90°, its effectiveness ranked lowest among the six training interventions in COD tests involving angles less than 90°. This discrepancy may be attributed to the reduced knee joint loading and significantly shorter braking times required during COD movements with smaller angles, as noted in line with Dos’Santos et al. (2018). The 90° may represent a critical threshold, beyond which COD tests become angle-dependent, requiring greater eccentric work by the knee joint during COD movement (Dos’Santos et al., 2018). When performing sharper COD, especially those greater than 90°, athletes undergo significant deceleration during the gait cycle prior to plant foot contact (Havens & Sigward, 2015b; Jones, Herrington & Graham-Smith, 2016a, 2016b; Jones et al., 2017; Nedergaard, Kersting & Lake, 2014).

For COD tests with angles less than 90°, our findings indicate that COM and CODT are the most effective interventions for improving performance. This result is encouraging because, in actual matches of sports such as football or rugby, most of the small-angle COD performed by athletes fall within this range (Praça et al., 2020). However, COM encompasses various types of training approaches (i.e., combined balance training and PT (Bouteraa et al., 2020), combined PT and CODT (Beato et al., 2018)), making it difficult to standardize. Any combination of two training modalities can be described as COM. Notably, our analysis included only two studies employing COM as an intervention: one combining RST and PT, and the other combining EOT and vibration training. A substantial body of evidence indicates that COM can enhance the health benefits of individual training modalities compared to relying solely on a single training program (Bouteraa et al., 2020; Hammami et al., 2019c; Makhlouf et al., 2018). However, this advantage does not extend to the well-documented incompatibility between endurance and strength training (Nader, 2006). This synergistic effect may explain the effectiveness of COM in improving COD performance with smaller angles. However, due to the limited number of studies, we are unable to further refine the results. Future research on different COM training interventions is needed, as this may help to maximize COD performance for angles less than 90°. However, given the limited number of studies, these findings should be interpreted with caution.

CODT, as one of the main power training modes, has been shown to significantly improve the performance of COD with angles under 90°. Compared to other training methods, such as TRT or EOT, CODT is more similar to the actual test. There is ongoing debate in the field regarding the effectiveness of exercises performed bilaterally in the vertical direction (e.g., weightlifting, squats, deadlifts), which appear to have limited impact on improving COD performance (Brughelli et al., 2008). In contrast, CODT, which closely resembles the COD test, may be a more effective way to enhance test performance. It could improve COD performance in two ways: first, during CODT, the technical execution of COD movements improves (Sheppard & Young, 2006). As adolescents may not have fully developed their technical skills, repeated CODT may enhance their technique; second, CODT is a horizontal exercise, aligning with the abilities required for the COD test, and these abilities are developed through CODT (Brughelli et al., 2008). Therefore, we suggest that for larger-angle COD, lower limb strength may be crucial. However, for smaller-angle COD, power output or horizontal movement may play a more significant role.

This review has several limitations. First, the included COD tests varied considerably, with some assessing only a single COD, while others, such as the Illinois agility test, included up to ten CODs. Tests with fewer CODs may place less emphasis on COD performance, making sprinting ability a more significant determinant of overall performance.

Second, despite our comprehensive search, there is still a lack of studies focusing on improving COD performance through training interventions in female team-sport adolescent athletes. Both the angle of COD and the sex of participants are important factors influencing COD performance. For example, the Illinois agility test, which incorporates both angles greater and less than 90°, has shown a moderate correlation with 20 m sprint time (r = 0.472). Similarly, Paul, Gabbett & Nassis (2016) found a strong correlation between T-test performance and 40-yard dash times in female athletes (r = 0.73), compared to a moderate correlation in males (r = 0.55). Thus, our findings should therefore be interpreted cautiously, as the studies involving COD angles less than 90° did not include female adolescent team-sport athletes, potentially limiting the generalizability of our results to female populations. Therefore, more research is needed to explore intervention methods aimed at enhancing COD performance in female athletes. Additional research is needed to confirm the effectiveness of COM and PT for improving COD performance at angles less than 90° in adolescent female team-sport athletes.

Third, the limited number of included studies constrained our analysis. For tests with COD angles less than 90°, five training interventions were not represented, and for angles greater than 90°, one intervention was missing. As a result, some conclusions relied on indirect comparisons, which lack direct evidence. Although these findings provide preliminary guidance for training, they require further validation through experimental studies focusing on different training interventions and their impact on adolescent team-sport athletes’ COD performance. Therefore, future studies should adopt high-quality experimental designs to provide more direct comparative evidence, clarifying the long-term adaptation of different training interventions.

Finally, the included studies covered a wide age range and involved participants with varying levels of biological maturity. While adolescents at different maturity stages generally demonstrate positive training adaptations, maturity is an important variable that can influence results. Future research should aim to develop more targeted training interventions tailored to the specific needs of adolescent team-sport athletes at different stages of maturity. Addressing these limitations will enable a deeper understanding of how to optimize COD performance through training interventions.

Conclusions

The current study highlights the effects of various training interventions on COD performance in adolescent team-sport athletes. When no restrictions were applied to the angle of COD tests, EOT emerged as the most effective intervention. Our findings further emphasize the importance of tailoring training interventions to the specific COD angles, distinguishing between angles below 90° and those exceeding 90°. Specifically, COM was identified as the most effective intervention for improving COD performance at angles below 90°, whereas EOT demonstrated superior efficacy for angles exceeding 90°. These findings underscore the need for angle-specific training strategies to optimize COD performance in this population. This result can provide coaches and strength trainers with insights for designing training programs that reduce the risk of ACL injuries during COD in adolescent team-sport athletes and enhance COD performance.

Supplemental Information

Supplemental Information 1 Search Strategy.

Supplemental Information 2 Risk of Bias.

Supplemental Information 3 Ranking of different interventions.

Supplemental Information 4 Evaluation of Heterogeneity and Inconsistency.

Supplemental Information 5 Funnel plot.

Supplemental Information 6 PRISMA 2020 checklist.

Supplemental Information 7 Rationale for systematic review and meta-analysis.

Additional Information and Declarations

Competing Interests

The authors declare that they have no competing interests.

Author Contributions

Yonghui Chen conceived and designed the experiments, performed the experiments, prepared figures and/or tables, authored or reviewed drafts of the article, and approved the final draft.

Maiwulanjiang Tulhongjiang conceived and designed the experiments, performed the experiments, authored or reviewed drafts of the article, and approved the final draft.

Tianpeng Ling analyzed the data, prepared figures and/or tables, and approved the final draft.

Xinmiao Feng performed the experiments, analyzed the data, prepared figures and/or tables, and approved the final draft.

Jing Mi analyzed the data, authored or reviewed drafts of the article, and approved the final draft.

Ruidong Liu performed the experiments, authored or reviewed drafts of the article, and approved the final draft.

Data Availability

The following information was supplied regarding data availability:

This is a systematic review/meta-analysis.

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
