# Peer review of "The optimal training intervention for improving the change of direction performance of adolescent team-sport athletes: a systematic review and network meta-analysis"

_PeerJ, doi:10.7717/peerj.18971_

## Round 0.1 · original submission · Major Revisions

· Academic Editor

Major Revisions

Dear Authors

Three experts in the study field have reviewed the manuscript. Your study has been reviewed with comprehensive comments to improve the quality of the manuscript. I would encourage you to highlight the study's novelty as the point raised by the reviewers. We invite you to submit a revised version of the manuscript addressing the reviewers’ comments.

We look forward to receiving your revised manuscript.

Best regards

Yung-Sheng Chen, Ph.D.
Academic Editor

Reviewer 1 ·

Basic reporting

no comment

Experimental design

no comment

Validity of the findings

no comment

Additional comments

Review manuscript ID: 107951v1

Title: The optimal training intervention for improving the change of direction performance of adolescent team-sport athletes: A systematic review and network meta-analysis

*Comments and Suggestions for Authors
*General comments
-The article “The optimal training intervention for improving the change of direction performance of adolescent team-sport athletes: A systematic review and network meta-analysis” aimed to synthesize the effects of diverse training interventions on COD performance. I greatly respect the effort made by the authors; however, I think that the major problem with this literature review is that it is a bit premature and lacks many recently published works about this topic. Indeed, I have a major concern about using this strategy as a principle and specific strategy for court-based sports players which is very general topic and need to be discussed by every discipline specific characteristic (which was not well developed and discussed in the present systematic review). Especially, that the authors didn’t well express and justified the importance of this work. Otherwise, from my modest expertise in the field I didn’t feel the novelty and the specificity of this work (Maybe every discipline needs to be addressed independently will give some fresh information’s to the field).
-The Introduction is very oriented to soccer more specifically and the title is about team sports in general (I believe that it still a lot of literature to add to this topic).
-With respect to the authors efforts, please you may explain and justify your new approaches in the introduction part (what is the novelty of this study compared to previous studies related to the topic)
*Specific comments
*Abstract
1) The background is very general.
2) the conclusion is very long (Please be precise with the most important findings)
*Introduction
1) Very general and does not represent the specificities of COD among Team sports players and the objective of the study need more justification.
*Methods
1) More recently published articles need to be included (the specific characteristics of every discipline needs to addressed and discussed in relation with COD).
*Results
1) This section needs to be more specific and well detailed based on every discipline characteristic in my modest opinion.

*Discussion
1) I advise the authors to avoid using long sentences... I suggest to reduce the length of this topic... It is very hard to follow for readers…
*Conclusion
1) The conclusions of this meta-analysis in my opinion can be considered with caution also concerning the practical applications considering the major limits mentioned. Also, this part is too long and there are repetitions of information already mentioned in the limits and practical application section.

Reviewer 2 ·

Basic reporting

Introduction
1. Theoretical framework (Lines 62-66): It would be beneficial to introduce a theoretical framework connecting eccentric training with improvements in change of direction (COD) performance. This would provide a solid foundation for the study's hypotheses.
2. Innovation in angles (Lines 75-79): Strengthen the justification for focusing on angles greater or less than 90°, highlighting how this perspective is novel and practically relevant.
3. Limitations of previous studies (Lines 84-90): Identify and discuss the limitations of prior studies related to COD. Consider incorporating the following relevant articles into the discussion:
o https://doi.org/10.23736/S0022-4707.20.10178-6
o https://doi.org/10.2165/00007256-200838120-00007
o https://doi.org/10.1007/s40279-018-0968-3
o https://doi.org/10.7752/jpes.2022.07215
o https://doi.org/10.1123/ijspp.2015-0694
4. Clear focus for the introduction (Lines 91-95): Limit this paragraph to the study’s objectives and the rationale for conducting this systematic review and network meta-analysis (NMA). Avoid including information derived from the methods section.
5. Female representation (Lines 318-319): If studies with female participants are included, it is necessary to highlight the low representation of women in previous research and how this constitutes a gap in the literature.

Experimental design

Materials and Methods
1. Multilingual strategy (Lines 101-108): Provide additional details on how multilingual studies were included, ensuring that all search terms are explicitly listed (Lines 104-105).
2. Justification of exclusions:
o Explain why patients with chronic instability were excluded (Lines 126-127).
o Justify the exclusion of studies lasting less than 4 weeks (Lines 117-118).
3. Participant selection (Lines 115-116): Clarify why participants aged 11-18 were chosen and how this selection impacts the generalisability of the results.

Results
1. Numerical inconsistency (Lines 209 and 177): Address the discrepancy between the number of studies reported in these lines (10 vs. 11 studies) and correct it to ensure consistency in the manuscript.

Validity of the findings

Discussion
1. Duration of interventions (Lines 257-260): Expand on the influence of intervention duration on the results, as one of the inclusion criteria was a minimum of 4 weeks. Evaluate how the length of the intervention affects outcomes and propose studies to explore longer durations.
2. Acknowledgement of uncertainty (Lines 262-263): Recognise the uncertainty inherent in using indirect results and propose practical solutions, such as the need for direct comparisons in future research.
3. Longitudinal studies (Lines 320-321): Propose future investigations examining the long-term effects of the different interventions analysed.

Additional comments

General Comments
1. Figure 5: The quality of Figure 5 needs to be improved to enhance its readability. Currently, the details are not easily visible, which may hinder the comprehension of the results.
2. Interpretation of Figures 5B and 5C: Upon analysing these figures, it is evident that most comparisons are based on only 3-4 studies. This significantly limits the strength of the conclusions. While this is acknowledged in the limitations, it would be helpful to reiterate this issue throughout the discussion to emphasise the lack of robustness in some findings.
3. Language review: It is recommended that the manuscript be reviewed by a native English speaker with expertise in the field. Some terms used are not appropriate in a scientific context and may reduce the clarity of the text.
4. Descriptive data table: While Table 2 is mentioned, it is not visible in the document. If it is not included, it is essential to add it, as it would provide a comprehensive overview of the studies analysed. Additionally, considering that female representation accounts for less than 20% of the total sample, the inclusion of these studies should be further justified. Would it be more appropriate to conduct a specific meta-analysis for women in the future and focus exclusively on the male population in this study to enhance the manuscript’s quality?
5. Use of acronyms: The text contains an excessive number of acronyms, which may complicate readability for a broader audience. It is recommended to reduce their use and ensure that each acronym is defined the first time it appears (e.g., in lines 228 and 270).
6. Errors in “adolescent”: The term "adolescent" is misspelled in several instances. Please review the manuscript to correct these errors.

Reviewer 3 ·

Basic reporting

The manuscript needs some work with regards to the writing. In parts it is written in past tense, and others in future or present. Please go through your manuscript and amend so it reads in past tense throughout. There is also a lot of ‘casual writing’ throughout. Please re-read manuscript and amend so it reads more formal.

Additionally, there are many cases where you are missing spaces between words and references, please check your manuscript thoroughly and amend (I have also mentioned this in my general comments).

I think the methods section needs to be written better, specifically the data analysis section, it feels ‘clunky’ and doesn’t read easily. You need to make it clear to the reader how you will analyze the data and why you are using these specific methods/graphs etc., The reader should be able to read your methods section and easily replicate what you did.

Many of the Figures are blurry and pixelated, this should be amended before publication in a journal. The order of the Figures mentioned in the text is not in ascending order, this needs to be amended. The figures need to have definitions for the types of training included at the bottom. Currently they are very hard to interpret.

I could not find the definition table or the training mode and currently it is hard to interpret the results section and what training methods you are talking about.

Experimental design

I do have some issues with the experimental design. Firstly, you have chosen the age range 9-17 years old, do you not think that maturation will play a role in how well the participants are responding to the type of training? Do you think it is appropriate to be comparing the results of a 9-year-old to the results of a 17 year old? I also think the rationale outlined in the introduction could be better regarding the age and angle. Why specifically youth athletes? Your rationale was to prevent slow or declining improvements in athletes COD performance due to age-related factors, but I would argue that athletes have not reached their peak in terms of COD performance by the age of 17-years old? I can see the knowledge gap, but feel the written rationale could do with some work.

You have categorized into angles below and above 90-degrees, however I would argue you could potentially have three categories as 90-degrees and 180-degrees are considered the same category but have different biomechanical and neuromuscular demands. Perhaps shallow cuts (1-50 degrees), Cuts (50-90-degrees) and turns (100-180-degrees) or something of the like? Let me know your thoughts.

It appears majority of your tables detailing the training modalities and studies included are missing? However perhaps you have missed some studies in the review, or you have reasons for not including the following:

Beato, M., Bianchi, M., Coratella, G., Merlini, M., & Drust, B. (2018). Effects of plyometric and directional training on speed and jump performance in elite youth soccer players. The Journal of Strength & Conditioning Research, 32(2), 289-296

Dos’ Santos, T., McBurnie, A., Comfort, P., & Jones, P. A. (2019). The effects of six-weeks change of direction speed and technique modification training on cutting performance and movement quality in male youth soccer players. Sports, 7(9), 205.

Sariati, D., Hammami, R., Zouhal, H., Clark, C. C., Nebigh, A., Chtara, M., ... & Ounis, O. B. (2021). Improvement of physical performance following a 6 week change-of-direction training program in elite youth soccer players of different maturity levels. Frontiers in Physiology, 12, 668437.

Michailidis, Y., Tabouris, A., & Metaxas, T. (2019). Effects of plyometric and directional training on physical fitness parameters in youth soccer players. International Journal of Sports Physiology and Performance, 14(3), 392-398.

These are just a few of the key ones that I couldn’t see included in this review. I found these just by doing a simple google scholar search. I think you also should include Google scholar in your database search, as this could be the reason why you have missed these studies. Please let me know if there is a reason you did not include.

Validity of the findings

As I have mentioned previously, I feel the rationale could be strengthened throughout the manuscript.

Also mentioned above is the absence of some studies. The authors should thoroughly check they have included all relevant literature or stated why or why not these were not included. These were found just from a brief search, therefore there could be several more missing articles.

I think you need to state a very clear and simple research question in your introduction and relate your findings back to this throughout your discussion.

Additional comments

Thank you for the work you are completing in the change of direction space! I appreciate the effort that has gone into the writing of this manuscript; however, I feel there are definitely areas that can be improved and should be done prior to the publication of this article. You can see my general and specific comments below pertaining to each section.

Introduction

Line 52: This sentence is a bit casual and could be written better e.g., Elite soccer athletes perform approximately 700 COD per match, with majority of these (~600) being between 1-90° (I wouldn’t say 0°, as this then would not be a COD).

Line 55: Space between direction and the reference needs to be added.

Line 55: Perhaps here you could offer specific examples of how having good COD provides a competitive advantage e.g., evasion of opponents, faster transitions from attack to defense etc.,

Line 56: Space between end of sentence and reference (this appears to occur throughout and should be amended.

Line 57: What is meant by ‘high-standard’ COD performances? I would say what puts athlete at a high risk of injury during these directional changes, is the deceleration forces experienced in order to change the momentum of the body.

Line 68: Why is age crucial? You state there is a decline in COD performance with increasing age. But that is implying the younger we are, for example 14 years, the better we are at COD. There will be a peak in an athletes life where they are performing at their best. This point in time will also depend on the sport, the journey of the athlete e.g., whether they have a high training age, early specialization etc.,

Look at the review by Ryan et al., 2022 where they reported weighted averages across a range of different studies that reported on 5-0-5 assessment. Based on their findings age 16-19 performed best for the traditional 5-0-5, however the over 20 group was best at the modified 5-0-5 test.

Ryan, C., Uthoff, A., McKenzie, C., & Cronin, J. (2022). Traditional and modified 5-0-5 change of direction test: Normative and reliability analysis. Strength & Conditioning Journal, 44(4), 22-37.

Line 77: You should mentioned the angle-velocity trade-off here, which is what you are alluding too.

Line 83: Removed the full stop after Falch reference and combine the sentences.

Line 87-88: Interesting sentence regarding the prior studies saying EOT is more effective for higher directional changes whereas TRT is preferred for cutting angles, however above in a previous sentence you say that greater than 90-degree is very force dominant and less than 90-degrees is speed-dominant, so why would TRT which is performed generally at slow velocities be better for COD less than 90-degrees?

Material and Methods:

Line 105: What are the * symbols for in ‘adolescen*’ and ‘team-sport athlete*’, also missing the t on adolescent.

Line 112: Change to ‘All included studies were’ ensure you are writing in past tense.

Line 113: Change to ‘there were no language restrictions’. What does this mean? You included all articles from any language?

Line 115: Were you focusing on males and females? Do you think we should be grouping males and females and generlising results to both?

Line 118: What was your rationale for a minimum of 4-weeks? Do you have a reference for this?

Line 119: You refer to Table 1 with definitions, however I can’t see this table in your included material?

Line 126: I recommend changing this to uninjured athletes.

Line 128-129: I’m not sure what you mean by this? Could you please explain.

Line 145: Change to (95% CI)

Line 149: Change to ‘All statistical analysis was performed using R…)

Line 155:-158: This can be worded better, for example ‘The I2 statistics was used as a measure of heterogeneity between studies. The I2 values ranged from 0-100% and were interpreted as follows: <25% low heterogeneity, 25-50% moderate heterogeneity and >75% high heterogeneity.’ Something like that.

In general, I feel this section is hard to read. It could be written to flow a lot better and easier to understand. This section needs to be written as if the reader has no understanding of what you have done and wants to follow your methods step by step. Please go through and try to re-write.


Results:

Again, this section is very ‘casually’ written and I think needs to be amended to read more professionally. Some specific examples can be seen below.


Line 168: Remove ‘According to the predetermined search strategy’ and reword.


Line 169-172: This needs to be re-written to read more professionally e.g.. ‘Initially, titles and abstracts were screened for relevance and 958 articles were excluded. The remaining articles (n = 1111) full text were screened and assessed for relevance, resulting in the inclusions of 37 studies. There was a total of 1257 participants included in the analysis with 80.80% being males (n = 1001) and 19.20% females (n = 238). One study did not report participant gender (REF), however was still included in the review because (insert your rationale).

Line 176: You refer to Table 2, but I can’t seem to see this in your attachments?

Line 178: You say 3-arm and 4-arm design, but you haven’t specified what this means and which studies it was.


Discussion:

Line 226: Change to ‘To the authors knowledge…’

Line 227: Removed ‘our findings’ and reword
Line 226-234: I think it could be helpful for your own writing if you come up with your main 3-4 findings and then base your subsequent paragraphs on those main findings. It is a bit hard to follow at the moment. For example: The main findings of the review are, 1) Out of the 12 intervention types, EOT appeared to be the most effective for enhancing COD performance, 2) With regards to cutting angles (<90-degrees), EOT, PT and TRT had similar effects on COD performance, and 3) for angles greater than 90-degrees, all five training modalities (XXXXX) significantly improved COD performance in youth team sport athletes.

Your three main discussion paragraphs will then be discussing those key findings.


Line 235: Perhaps you could expand on why eccentric strength is considered a crucial element? I think a key reason eccentric training has such an impact on COD performance, specifically in the younger population, is the increased ability to decelerate effectively, especially during the greater angles of directional change.

Line 266: Can you specify what equipment was used during the EOT?

Line 267: Change to Falch and colleagues have stated that…

Line 268: Fix the reference, do not include first name

Line 268: Try to refrain from using the word our. You have done this throughout, please go through and amend. For example you could say ‘The sub-group analysis in the current study…’

Line 286: The closing sentence, can you really say that? Since EOT had the greater impact?

Line 299-300: Im struggling to find the connection to chronic improvements in COD? How do you think PAPE relates to chronic improvements in COD performance? If you are going to provide examples like you have on line 300, I think it needs to be specific to COD performance, jumping is not relevant in this context.

Line 301: Again, you need to link everything back to COD performance and why any of this is important in terms of enhancing COD in youth.

Line 304: I am not sure what you mean here in terms of the load stimulus during deceleration increases? Can you please explain this.


Line 308: You say that most of the studies included sprints or jumps, was sprints the ones that saw the most improvements? I would just say that sprinting and the combination of lifting (you explained how this can benefit COD in previous paragraphs) is the explanation for the improvement. It is likely if they are doing sprints, they are improving their accelerative and reaccelerative ability, which makes up ~50% of a 180-degree COD test (see the work of Ryan and colleagues, 2022).

Ryan, C., Uthoff, A., McKenzie, C., & Cronin, J. (2022). Profiling change of direction ability using sub-phase 5-0-5 analysis. International Journal of Strength and Conditioning, 2(1).


Line 310-311: I can understand how strength training and PAPE can have an acute effect on COD performance, but struggling to see how you are linking this to chronic adaptation, which is what your review is about.

Line 312: what do you mean by COD-specific neural muscular strength training?


Line 314: Limitations section, do you also think you could discuss the large age range and different maturation statuses and how these could impact results? In an ideal world it really should be categorized also by biological age.

Conclusions:

You need to have a strong question in your introduction and link back to this here and how you answered it through the completion of your review.

Line 325: Reword to something like the following: ‘The current study details the effects of different training interventions on COD performance in adolescent team-sport athletes.”

Line 327: Change to “It appears that EOT is the most effective training modality, irrespective of the COD angle performed.”


Reference list:

Please go through your reference list and check for consistency e.g., all journals should be capitalised, consistent use of ‘and’ or ‘&’, consistent use of journal abbreviations (majority you have written out the full journal name, so go ahead and change the abbreviated ones to full names).

---

## Round 0.2 · accepted · Accept

· Academic Editor

Accept

Dear Authors,

I would like to express my big heart for your patience and efforts to improve the quality of the manuscript. Your submission is now endorsed by two experts for acceptance of publication in PeerJ. Congratulations!!!

Thank you for submitting your article to PeerJ. I look forward to receiving your research and review articles in the future.

Best Regards
Ph.D. Yung-Sheng Chen

Reviewer 1 ·

Basic reporting

no comments

Experimental design

no comments

Validity of the findings

no comments

Additional comments

The authors have done a good job and the article has improved appropriately. Thank you

Reviewer 2 ·

Basic reporting

The manuscript has improved notably. The author has expended considerable effort in revising the manuscript, and I accept it in its present form.

Experimental design

The manuscript has improved notably. The author has expended considerable effort in revising the manuscript, and I accept it in its present form.

Validity of the findings

The manuscript has improved notably. The author has expended considerable effort in revising the manuscript, and I accept it in its present form.